# TMC7 deficiency causes acrosome biogenesis defects and male infertility in mice

Jing Wang[1,2], Yingying Yin[1,3], Lei Yang[1,2], Junchao Qin[1,2], Zixiang Wang[1,2], Chunhong Qiu[1], Yuan Gao[3], Gang Lu[4], Fei Gao[5], Zi-Jiang Chen[3], Xiyu Zhang[1]*, Hongbin Liu[3]*, Zhaojian Liu[1,2]*

[1]Key Laboratory of Experimental Teratology, Ministry of Education, Department of Cell Biology, School of Basic Medical Sciences, Shandong University, Jinan, China; [2]Advanced Medical Research Institute, Shandong University, Jinan, China; [3]Center for Reproductive Medicine, Shandong University, Jinan, China; [4]CUHK-SDU Joint Laboratory on Reproductive Genetics, School of Biomedical Sciences, The Chinese University of Hong Kong, Hong Kong, China; [5]State Key Laboratory of Stem Cell and Reproductive Biology, Institute of Zoology, Chinese Academy of Sciences, Beijing, China

**\*For correspondence:**
xiyuzhang@sdu.edu.cn (XZ);
hongbin_sduivf@aliyun.com (HL);
liujian9782@sdu.edu.cn (ZL)

**Competing interest:** The authors declare that no competing interests exist.

**Abstract** Transmembrane channel-like (TMC) proteins are a highly conserved ion channel family consisting of eight members (TMC1–TMC8) in mammals. TMC1/2 are components of the mechanotransduction channel in hair cells, and mutations of TMC1/2 cause deafness in humans and mice. However, the physiological roles of other TMC proteins remain largely unknown. Here, we show that *Tmc7* is specifically expressed in the testis and that it is required for acrosome biogenesis during spermatogenesis. *Tmc7*$^{-/-}$ mice exhibited abnormal sperm head, disorganized mitochondrial sheaths, and reduced number of elongating spermatids, similar to human oligo-astheno-teratozoospermia. We further demonstrate that TMC7 is colocalized with GM130 at the *cis*-Golgi region in round spermatids. TMC7 deficiency leads to aberrant Golgi morphology and impaired fusion of Golgi-derived vesicles to the developing acrosome. Moreover, upon loss of TMC7 intracellular ion homeostasis is impaired and ROS levels are increased, which in turn causes Golgi and endoplasmic reticulum stress. Taken together, these results suggest that TMC7 is required to maintain pH and ion homeostasis, which is needed for acrosome biogenesis. Our findings unveil a novel role for TMC7 in acrosome biogenesis during spermiogenesis.

## eLife assessment

This study reports an **important** discovery highlighting the essential role of the putative ion channel, TMC7, in acrosome formation during sperm development and thus male fertility. The evidence for the requirement of TMC7 in acrosome biogenesis and sperm function is **convincing**, although its function as an ion channel remains to be further determined. Overall, this work will be of great interest to developmental biologists and ion channel physiologists alike.

## Introduction

Spermiogenesis is the post-meiotic phase of male germ cell development during which the haploid spermatids differentiate into mature spermatozoa (*Kang et al., 2022*). Mouse spermiogenesis consists of 16 steps characterized by morphological changes in acrosome formation, nuclear condensation and

elongation, flagellum formation, and cytoplasm removal (*Li et al., 2014*). Acrosome biogenesis is one of the key events of spermiogenesis, and it begins with the initial stage of spermatid development. The acrosome is a specialized organelle with a cap-like structure covering the anterior part of the mature spermatozoon head and is indispensable for the fertilization process (*Xiong et al., 2021*). Although acrosome biogenesis is thought to be derived from the Golgi complex in early spermatids, growing evidence suggests that the endoplasmic reticulum (ER) and lysosomes are involved in acrosome biogenesis (*Khawar et al., 2019*). Defects in acrosome biogenesis can result in subfertility or infertility.

Acrosome biogenesis is generally classified into four successive phases, namely the Golgi, cap, acrosome, and maturation phases (*Xu et al., 2023*). A number of ER and Golgi-associated proteins have been shown to be involved in acrosome biogenesis. Heat shock protein 90B1 (HSP90B1) is an ER molecular chaperone that has been shown to be a testis-specific protein, and HSP90B1 deficiency leads to a globozoospermia-syndrome like phenotype (*Audouard and Christians, 2011*). GM130 (Golgi matrix protein 130) localize on the *cis* surface of the Golgi apparatus, and loss of GM130 causes acrosome malformations and results in male infertility (*Han et al., 2017*). GOPC (Golgi-associated PDZ and coiled-coil motif-containing) is a *trans*-Golgi-associated protein that plays a role in vesicle transport from the Golgi apparatus. GOPC deficiency leads to vesicle transport failure, which in turn results in malformation of the acrosome and to infertility (*Yao et al., 2002*). Other vesicle transport proteins including PICK1, ICA1l, and VPS13b are involved in proacrosomal vesicle trafficking during acrosome formation (*Xiao et al., 2009*; *He et al., 2015*). Recent evidence suggests that autophagy is involved in proacrosomal vesicle fusion and transport to form the acrosome, and disruption of ATG7 (*Wang et al., 2014*; *Huang et al., 2021*) or ATG5 causes aberrant acrosome formation and male infertility or subfertility. Despite these findings, further studies are needed to better understand the complexity of acrosome biogenesis.

Transmembrane channel-like (TMC) proteins are highly conserved ion channel-like proteins, with eight family members (TMC1–TMC8) identified in humans and mice (*Keresztes et al., 2003*; *Kurima et al., 2003*; *Yue et al., 2019*; *Kurima et al., 2002*). TMC proteins are predicted to have 6–10 transmembrane domains, and all protein members contain a conserved 120 amino acid TMC domain (*Yue et al., 2019*). The single-particle cryo-electron microscopy structure shows that TMC1 assembles as dimers and that each monomer contains 10 transmembrane domains with four domains (S4–S7) that line the channel pore (*Jeong et al., 2022*). Tmc1 and 2 are mechano-electrical transducer (MET) channels in cochlear hair cells, and mutations in *Tmc1/2* cause deafness in humans and mice (*Nist-Lund et al., 2019a*) by disrupting hair cell mechanoelectrical transduction, and *Tmc* gene therapy can restore auditory function and balance in mice (*Askew et al., 2015*; *Nist-Lund et al., 2019b*). Germline mutations in *Tmc6* and *Tmc8* are associated with epidermodysplasia verruciformis, which leads to increased susceptibility to human papilloma virus (HPV) infection (*Kurima et al., 2003*; *Yue et al., 2019*). Among the eight TMC protein members, TMC1 and TMC2 are well studied due to their role in hearing conduction, and understanding the physiological and functional roles of other TMC family members remains a major challenge.

In this study, we found that *Tmc7* begins to be expressed in diplotene spermatocytes and that it is expressed at high levels in round spermatids in mouse testes. By generating *Tmc7* conventional knockout mice (*Tmc7*$^{-/-}$), we demonstrate that TMC7 deficiency leads to aberrant Golgi morphology and impaired fusion of Golgi-derived vesicles to the acrosome, thus resulting in male infertility with an oligo-astheno-teratozoospermia (OAT) phenotype. Our findings provide new insights into the physiological role of TMC family members in spermiogenesis.

## Results

### *Tmc7* expression is confined to the testis, and deletion of *Tmc7* leads to male infertility

To explore the roles of *Tmc* family members during spermatogenesis, we analyzed the expression of *Tmc1–Tmc8* using the mouse developmental transcriptome (E-MTAB-6798) (*Cardoso-Moreira et al., 2019*). Among the eight *Tmc* family members, we found that only *Tmc5* and *Tmc7* exhibited elevated expression in the testis during development (*Figure 1A*). *Tmc7* caught our great interest because *Tmc7* was expressed at higher levels than *Tmc5* in the testis during development. We next

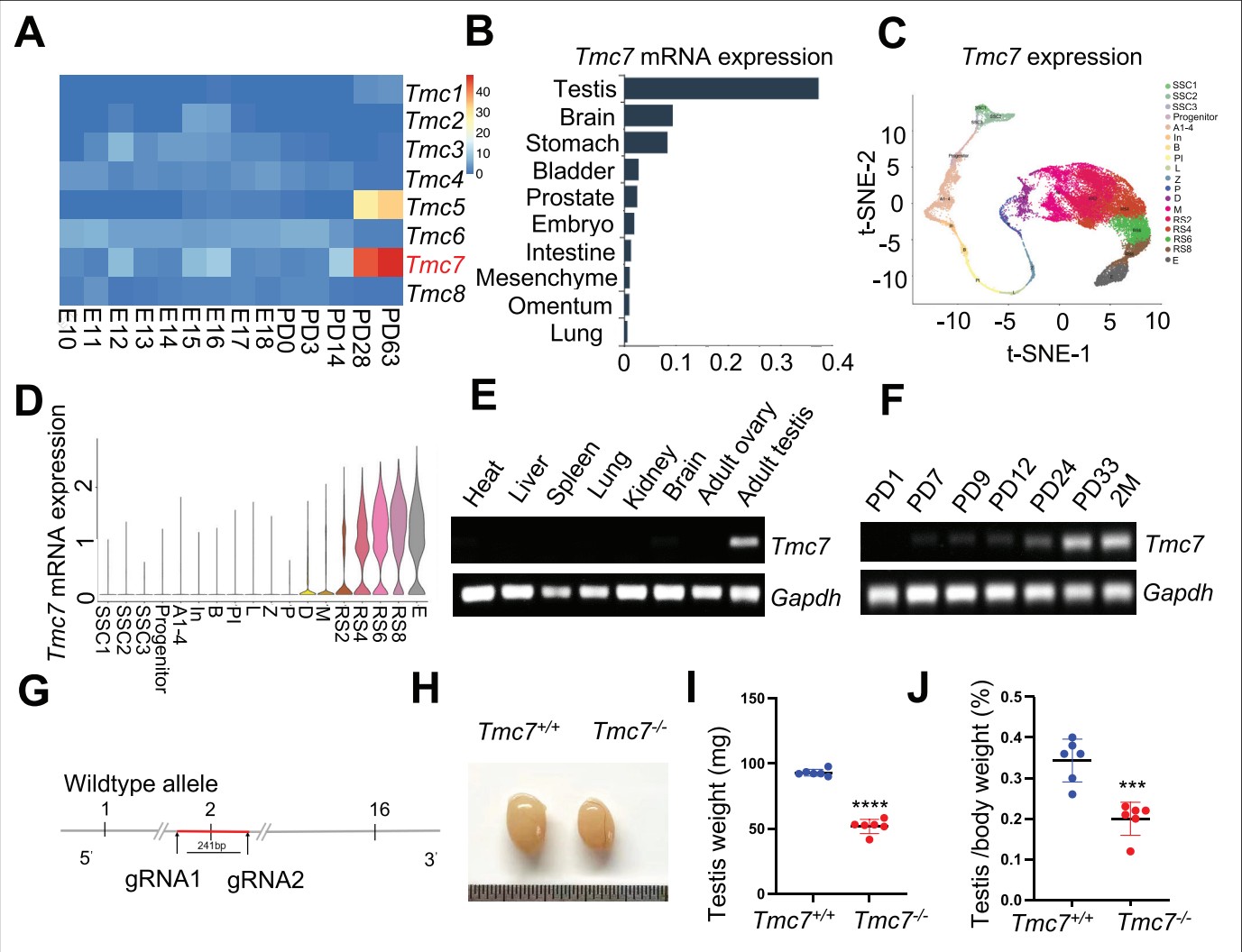

**Figure 1.** *Tmc7* expression is confined to the testis, and its deletion leads to male infertility. (**A**) Heatmap analysis of the expression of *Tmc* family members at embryonic day (E) 10–E18 and different postnatal days (P0, P3, PD14, PD28, and PD63) in mouse testis using the Evo-devo mammalian organ database (http://apps.Kaessmannlab.org/evodevoapp/). (**B**) The analysis of *Tmc7* expression in different tissues using the Mouse Cell Atlas database (https://bis.zju.edu.cn/MCA/). (**C, D**) *t*-Distributed Stochastic Neighbor Embedding (*t*-SNE) plot and violin plot showing the expression of *Tmc7* in different mouse germ cells using the Male Health Atlas (http://malehealthatlas.cn/). SSC1-3: spermatogonial stem cells I–III; Progenitor: spermatogonia progenitor cell; A1–4: A1–4 spermatogonia; In: intermediate spermatogonia; B: type B spermatogonia; PI: preleptotene; L: leptotene; Z: zygotene; P: pachytene; D: diplotene; M: metaphase; RS: round spermatid. E: elongating spermatid. (**E**) The expression of *Tmc7* in different adult mouse tissues was measured by RT-PCR, including the heart, liver, spleen, lung, kidney, brain, ovary, and testis. *Gapdh* was used as the control. (**F**) The expression of *Tmc7* at different postnatal times (PD1, PD7, PD9, PD12, PD24, PD33, and 2 months) was measured by RT-PCR. *Gapdh* was used as the control. (**G**) Schematic illustrating the generation of *Tmc7*−/− mice using CRISPR/Cas9. (**H**) Testis from 9-week-old WT and *Tmc7*−/− mice. (**I**) Testicular weights of 9-week-old WT and *Tmc7*−/− mice. Six adult mice of each genotype were used. (**J**) Testis/body weight ratio of 9-week-old WT and *Tmc7*−/− mice. Six mice of each genotype were used. Images are representative of at least three independent experiments. The results are presented as the mean ± S.D., and two-sided unpaired Student's *t*-tests were used to calculate the p-values (*p<0.05, **p<0.01, ***p<0.001).

The online version of this article includes the following source data and figure supplement(s) for figure 1:

**Source data 1.** Uncropped and labeled gels for *Figure 1*.

**Figure supplement 1.** Phenotypic characterization of *Tmc7*−/− mice.

**Figure supplement 1—source data 1.** Uncropped and labeled gels for *Figure 1—figure supplement 1*.

**Figure supplement 1—source data 2.** Raw unedited gels for *Figure 1—figure supplement 1*.

analyzed *Tmc7* expression in different tissues using single-cell transcriptomic data from the Mouse Cell Atlas database (*Han et al., 2018*). We found that *Tmc7* was predominantly expressed in the testis compared with all other analyzed tissues (*Figure 1B*). Further, we analyzed *Tmc7* in different stages of germ cell development using single-cell sequencing data of male-specific tissues and organs in mice and humans (*Zhao et al., 2023*). Interestingly, *Tmc7* began to be expressed in diplotene spermatocytes and was expressed at high levels in round spermatids in mouse testes (*Figure 1C and D*). The *Tmc7* expression pattern was confirmed by RT-PCR (*Figure 1E and F*), and *Tmc7* was again found to be highly expressed in round spermatids of human testis (*Figure 1—figure supplement 1*). The distinct expression pattern in the testis indicates the potential role of *Tmc7* in spermatogenesis.

We generated *Tmc7*-deficient mice (*Tmc7*⁻/⁻) by partial deletion of exon 2 (the first 241 bp) using the CRISPR/Cas9 system (*Figure 1G*). *Tmc7*⁻/⁻ mice were confirmed by PCR-based genotyping and immunoblotting of the separated cytoplasmic and membrane testis proteins (*Figure 1—figure supplement 1*). We then examined the phenotypes and found that *Tmc7*⁻/⁻ mice were viable and had a normal appearance and exhibited a normal metabolic rate (*Figure 1—figure supplement 1*). However, *Tmc7*⁻/⁻ male mice showed smaller testes (*Figure 1H1*) and a reduced testis/body weight ratio (*Figure 1J*) compared with control mice. Moreover, *Tmc7*⁻/⁻ male mice exhibited complete infertility (*Supplementary file 1*). No difference in the number of tubules per testicle or in the number of Sertoli cells was seen when comparing *Tmc7*⁻/⁻ and WT mice; Sertoli cell were identified by immunofluorescence staining of the Sertoli cell marker SOX9 (*Figure 1—figure supplement 1*). These findings suggest that TMC7 is essential for spermatogenesis and thus for male fertility.

## TMC7 deficiency leads to severe defects in spermiogenesis

To determine the cause of infertility in *Tmc7*⁻/⁻ mice, we examined the composition of cell types in seminiferous tubules and cauda epididymis sections using hematoxylin staining in mouse testes at different developmental stages. In seminiferous tubules, the populations of germ cells were comparable between wild type (WT) and *Tmc7*⁻/⁻ mice at postnatal day (PD)14 and PD21. Intriguingly, the number of elongating spermatids in *Tmc7*⁻/⁻ mice decreased significantly compared with WT mice at PD28 (*Figure 2—figure supplement 1*). We next examined the testis and cauda epididymis sections of 9-week-old WT and *Tmc7*⁻/⁻ mice. We observed spermiogenesis defects in *Tmc7*⁻/⁻ mice as evidenced by the reduced number of elongating spermatids, the increased number of sloughed germ cells, and large vacuoles in the seminiferous tubules and cauda epididymis (*Figure 2A and B*). The vacuolation became more serious with increasing age (*Figure 2—figure supplement 1*). The total number of sperm in the cauda epididymis was drastically decreased in *Tmc7*⁻/⁻ mice (~0.15 × 10⁶ cells) compared to WT mice (~8.0 × 10⁶ cells) (*Figure 2C*). The morphological assessment of spermatozoa from the caudal epididymis revealed that a large proportion of spermatozoa (~70%) in *Tmc7*⁻/⁻ mice exhibited abnormal head morphology, similar to human OAT (*Figure 2D and E*). In parallel with this, periodic acid Schiff (PAS) staining indicated that a large number of spermatids failed to differentiate into elongated spermatids with sickle-shaped heads and instead exhibited abnormal head morphology. The total number of spermatids began to significantly decrease at steps 9–12 of spermiogenesis (*Figure 2F and G*), and we performed Mito-tracker staining and found shortened or disorganized mitochondrial sheaths in *Tmc7*⁻/⁻ spermatozoa collected from the cauda epididymis (*Figure 2H*). Meanwhile, no motile or progressive sperm were found in the *Tmc7*⁻/⁻ mice (*Figure 2I and J*). Moreover, meiotic progression was not affected in *Tmc7*⁻/⁻ mice as evidenced by immunofluorescence staining of the meiosis markers SYCP3 (red) and γH2AX (green) in the seminiferous tubules of PD20 WT and *Tmc7*⁻/⁻ mice (*Figure 2—figure supplement 2*). Overall, we concluded that TMC7 ablation results in defects in spermiogenesis and in OAT-like phenotypes.

## Acrosome biogenesis is impaired starting from the Golgi phase in *Tmc7*⁻/⁻ spermatids

Acrosome biogenesis is one of the key events of spermiogenesis. To investigate whether TMC7 is required for the proper formation of the acrosome, FITC-labeled peanut agglutinin (PNA) was used to assess the acrosomal status of spermatids in 9-week-old WT and *Tmc7*⁻/⁻ mice. The spermatids of WT mice had normal structures in the Golgi, cap, acrosome, and maturation phases, while spermatids in *Tmc7*⁻/⁻ mice exhibited abnormal acrosome morphology as early as the Golgi phase. Fragmented and malformed acrosomes were observed in all four phases of spermatids in the *Tmc7*⁻/⁻ mice (*Figure 3A*).

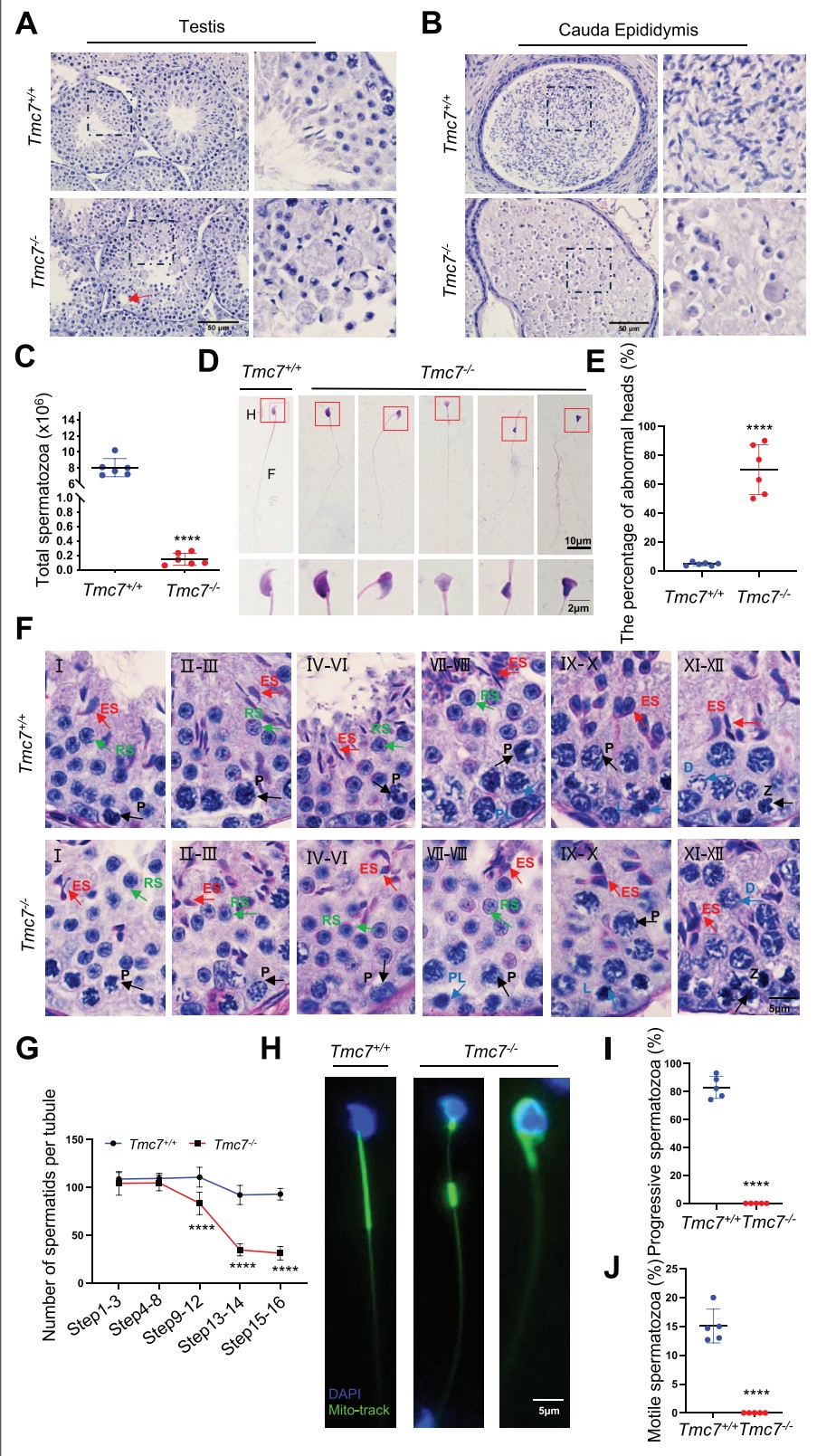

**Figure 2.** TMC7 deficiency leads to severe defects in spermiogenesis. (**A, B**) Hematoxylin staining in testis and caudal epididymis sections of 9-week-old WT and *Tmc7*[−/−] male mice. Scale bar, 50 μm. (**C**) The total number of sperm in the caudal epididymis from 9-week-old WT and *Tmc7*[−/−] male mice. Six mice were used. (**D**) Hematoxylin-eosin staining of the spermatozoa isolated from the caudal epididymis in 9-week-old WT and *Tmc7*[−/−] male mice.

*Figure 2 continued on next page*

*Figure 2 continued*

H: head, F: flagella. Scale bar, 10 μm, 2 μm. (**E**) The percentage of spermatozoa with abnormal heads in the caudal epididymis from WT and *Tmc7*$^{-/-}$ male mice. Six mice were used. (**F**) Periodic acid Schiff (PAS) and hematoxylin staining in 9-week-old WT and *Tmc7*$^{-/-}$ mouse seminiferous tubules. The numbers represent the stages of spermatogenesis. P: pachytene spermatocyte, PL: preleptotene spermatocyte, L: leptotene spermatocyte, Z: zygotene spermatocyte, D: diplotene spermatocyte, RS: round spermatid, ES: elongating spermatid. Scale bar, 5 μm. (**G**) Quantification of spermatid numbers in seminiferous tubules from spermatid stages 1–16. N = 15 tubules per stage and per genotype; three mice per genotype were used and five tubules per stage and per mouse were counted. (**H**) Mitochondrial sheath staining by Mito-tracker in sperm collected from the caudal epididymis. Scale bar, 5 μm. (**I, J**) The percent of progressive and motile spermatozoa in 9-week-old WT and *Tmc7*$^{-/-}$ mice. Five mice were used. Images are representative of at least three independent experiments. The results are presented as the mean ± S.D., and two-sided unpaired Student's *t*-tests were used to calculate the p-values (*p<0.05, **p<0.01, ***p<0.001).

The online version of this article includes the following figure supplement(s) for figure 2:

**Figure supplement 1.** Histological detection in WT and *Tmc7*$^{-/-}$ mice.

**Figure supplement 2.** TMC7 is not required for meiotic progression.

Absent or fragmented acrosomes were also found in *Tmc7*$^{-/-}$ spermatids collected from the cauda epididymis (*Figure 3—figure supplement 1*). To better understand the morphologic defects of the acrosome in greater detail, transmission electron microscopy (TEM) was performed to analyze the acrosome structure in WT and *Tmc7*$^{-/-}$ mice. We found that large vesicles accumulated adjacent to developing acrosomes in all phases of acrosome biogenesis, starting from the Golgi phase (*Figure 3B*, left panel) in the spermatids of *Tmc7*$^{-/-}$ mice. Additionally, abnormal acrosome structures and defective nuclear condensation were observed in *Tmc7*$^{-/-}$ spermatids (*Figure 3B*, right panels). Further investigation revealed that knockout of *Tmc7* resulted in Golgi apparatus fragmentation and disorganization, as determined by TEM and immunofluorescence staining of the Golgi marker GM130 (*Figure 3B and C*). The aspect ratio (height over width) of the Golgi apparatuses in *Tmc7*$^{-/-}$ spermatids (R = ~2.9) based on TEM images were also significantly smaller than in WT spermatids (R = ~4.9) (*Figure 3D*). Meanwhile, we found that the Golgi body failed to correctly orient toward the acrosome (*Figure 3—figure supplement 1*). We further performed immunoblotting to measure Golgi stress markers, HSP47 and TFE3 in 3- and 9-week-old *Tmc7*$^{-/-}$ testes compared with WT testes. The results showed that the protein levels of HSP47 and TFE3 were significantly increased upon loss of TMC7, indicating the occurrence of Golgi stress (*Figure 3E*, *Figure 3—figure supplement 1*). GOPC is localized in the *trans*-Golgi network and is involved in the transport and fusion of the proacrosomal vesicles with the acrosome (*Yao et al., 2002*). By immunostaining for GOPC and PNA, we found that GOPC staining (red) was colocalized with PNA (green) in WT spermatids. In contrast, colocalized GOPC and PNA spermatids were significantly decreased in *Tmc7*$^{-/-}$ mice (*Figure 3F*). Together, these findings suggest that TMC7 is required for proacrosomal vesicle trafficking and fusion during acrosome biogenesis.

## TMC7 localizes to the *cis*-Golgi, and this is required for maintaining Golgi integrity

To further delineate the role of TMC7 in acrosome formation, we next sought to determine the subcellular localization of TMC7 in the testis. We performed immunostaining and found that TMC7 was predominantly expressed in the perinuclear regions of pachytene spermatocytes and round spermatids in WT testes but not in *Tmc7*$^{-/-}$ testes, suggesting the specificity of the antibody (*Figure 4A*). By immunofluorescence co-staining of TMC7 with the *cis*-Golgi marker GM130, the *trans*-Golgi marker TGN46, and PNA, we found that TMC7 colocalized closely with GM130 but not with TGN46 (*Figure 4B and C*, *Figure 4—figure supplement 1*), indicating its localization to the *cis*-Golgi apparatus. Further evaluation showed that TMC7 and PNA were closely associated but not colocalized (*Figure 4D*, *Figure 4—figure supplement 1*). In parallel, we carried out co-immunoprecipitation with TMC7 antibody coupled with immunoblotting in mouse testes and found that TMC7 was able to pull down GM130, the GM130-interacting protein P115, and GRASP65 (*Figure 4E*), further confirming that TMC7 is localized to the *cis*-Golgi apparatus.

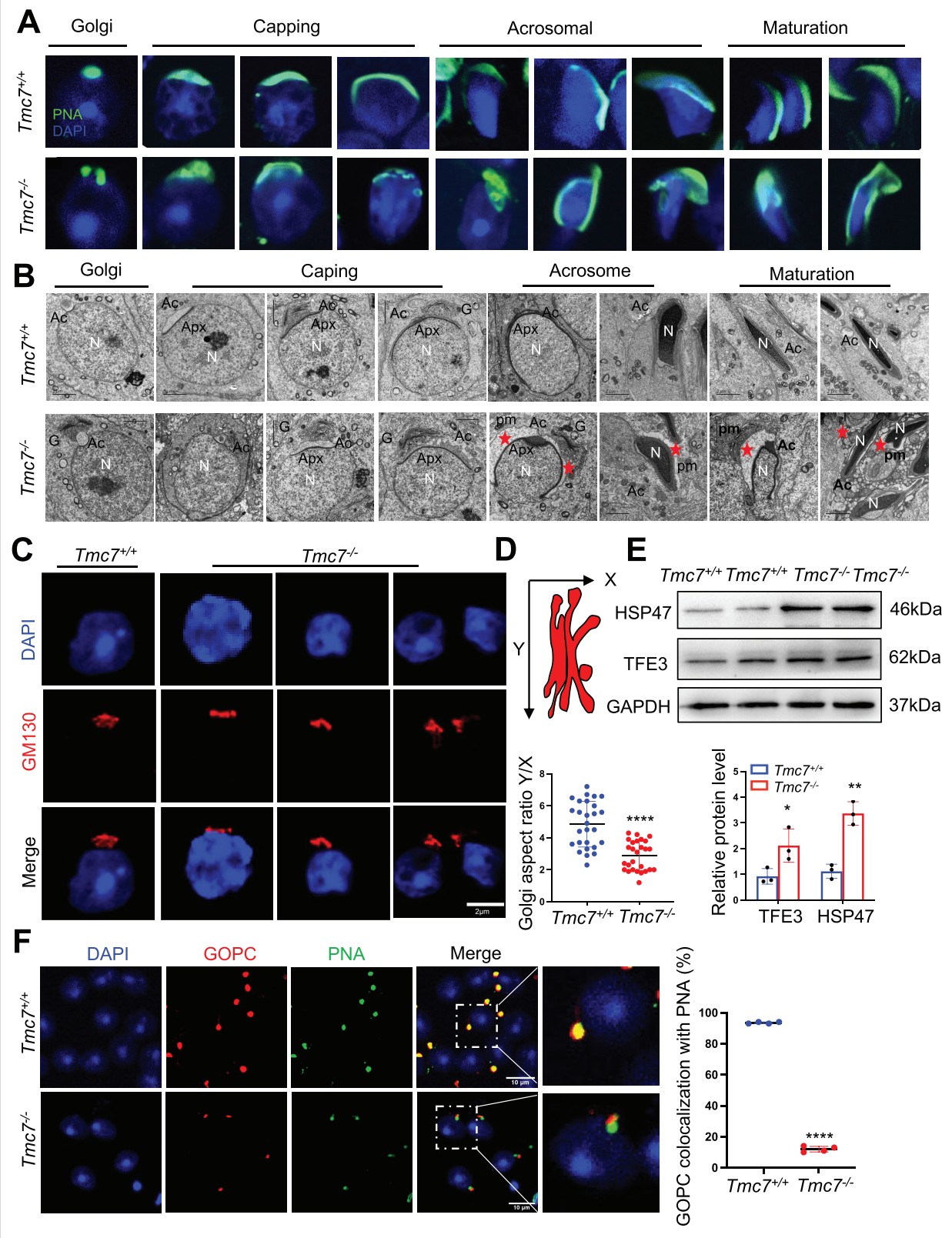

**Figure 3.** Acrosome biogenesis is impaired starting from the Golgi phase in *Tmc7⁻ᐟ⁻* spermatids. (**A**) Immunofluorescence staining of PNA (green) in the phases of acrosome biogenesis in 9-week-old WT and *Tmc7⁻ᐟ⁻* spermatids. Nuclei were stained with DAPI (blue). The phases of acrosome biogenesis and corresponding spermiogenesis steps are the Golgi phase (steps 1–3), cap phase (steps 4–7), acrosome phase (steps 8–12), and maturation phase (steps 13–16). (**B**) Transmission electron microscopy (TEM) images of testicular sections of 9-week-old WT and *Tmc7⁻ᐟ⁻* mice. The red star represents the

*Figure 3 continued on next page*

Figure 3 continued

abnormal plasma membrane space and accumulated vesicles. N: nucleus, Ac: acrosome; Apx: acroplaxome; pm: plasma membrane. G: Golgi complex; Scale bar, 2 µm. (**C**) Immunofluorescence staining with the antibody against GM130 in spermatids from the testes of 9-week-old WT and $Tmc7^{-/-}$ mice. Scale bar, 2 µm. (**D**) Calculation of the aspect ratio (height [X] over width [Y]) of 9-week-old WT and $Tmc7^{-/-}$ Golgi apparatuses based on TEM images. N = 27, and nine tubules in each testis were counted from three mice. (**E**) Western blot analysis of protein levels of Golgi stress-associated proteins (HSP47, TFE3) in 3-week-old WT and $Tmc7^{-/-}$ testes. ImageJ was used to quantify the protein level. (**F**) Immunofluorescence staining of GOPC (red) and PNA (green) and the quantitative analysis of GOPC and PNA lectin colocalization in squashed tubules of 9-week-old WT and $Tmc7^{-/-}$ mice. Nuclei were stained with DAPI (blue). Scale bar, 10 µm. Four mice were used, and 200 cells were counted per mouse. Images are representative of at least three independent experiments. The results are presented as the mean ± S.D., and two-sided unpaired Student's $t$-tests were used to calculate the p-values (*p<0.05, **p<0.01, ***p<0.001).

The online version of this article includes the following source data and figure supplement(s) for figure 3:

**Source data 1.** Uncropped and labeled gels for *Figure 3*.

**Source data 2.** Raw unedited gels for *Figure 3*.

**Figure supplement 1.** The defects of acrosome and Golgi apparatus in $Tmc7^{-/-}$ spermatozoa.

**Figure supplement 1—source data 1.** Uncropped and labeled gels for *Figure 3—figure supplement 1*.

**Figure supplement 1—source data 2.** Raw unedited gels for *Figure 3—figure supplement 1*.

GM130 is reported to be involved in maintaining the function and integrity of the Golgi apparatus (*Han et al., 2017*). To gain more insight into the role of TMC7 in the Golgi apparatus, we performed immunostaining against GM130, P115, and GRASP65 in the testes of 9-week-old WT and $Tmc7^{-/-}$ mice. The results showed that in the same stage seminiferous tubules, the protein levels of GM130, P115, and GRASP65 were significantly reduced in $Tmc7^{-/-}$ testes compared to WT testes (*Figure 4F and G*, *Figure 4—figure supplement 1*). Immunoblotting further verified the reduced levels of GM130, P115, and GRASP65 upon loss of TMC7 in the testes of 3- and 9-week-old mice (*Figure 4H*, *Figure 4—figure supplement 1*). These results suggest that TMC7 colocalizes to GM30 in the *cis*-Golgi and that this is required for maintaining Golgi integrity.

## TMC7 depletion impairs cellular homeostasis and leads to spermatid apoptosis

Intracellular ion homeostasis is essential for proper function, including membrane trafficking, protein glycosylation, and protein sorting (*Li and Wang, 2022*). The Golgi apparatus has been shown to be another intracellular $Ca^{2+}$ store in addition to the ER (*Lissandron et al., 2010*). $Tmc1$ mutation leads to reduced $Ca^{2+}$ permeability, thus triggering hair cell apoptosis and subsequent deafness (*Fettiplace et al., 2022*). We measured intracellular $Ca^{2+}$ by flow cytometry in WT and $Tmc7^{-/-}$ germ cells isolated from PD21 and PD30 testes. The results showed that the intracellular $Ca^{2+}$ concentration was significantly reduced in $Tmc7^{-/-}$ germ cells compared to WT germ cells (*Figure 5A*, *Figure 5—figure supplement 1*). In contrast, intracellular pH was increased in $Tmc7^{-/-}$ germ cells compared to WT germ cells (*Figure 5B*, *Figure 5—figure supplement 1*). These results suggest that the pH and ion homeostasis was impaired. We further measured the reactive oxygen species (ROS) levels and observed that ROS levels were significantly increased upon loss of TMC7 (*Figure 5C*, *Figure 5—figure supplement 1*) In addition, we examined ER stress-related proteins by immunoblotting and found that p-EIF2α, CHOP, and GRP78 were significantly increased in 3- and 9-week-old $Tmc7^{-/-}$ testes compared with WT testes, indicating the occurrence of ER stress (*Figure 5D*, *Figure 5—figure supplement 1*). Moreover, we carried out TUNEL staining in the seminiferous tubules and cauda epididymis. There was no difference in the number of TUNEL-positive cells between 3-week-old WT and $Tmc7^{-/-}$ mice of seminiferous tubules. The TUNEL-positive cells in seminiferous tubules and cauda epididymis were significantly increased in 9-week-old $Tmc7^{-/-}$ mice compared to WT mice (*Figure 5E*). The expression of apoptosis-related proteins was next measured by immunoblotting in 9-week-old WT and $Tmc7^{-/-}$ mice, depletion of TMC7 increased the levels of Cleaved-CASPASE-3 and BAX and decreased the levels of BCL2 (*Figure 5F*). These results support the hypothesis that TMC7 is required to maintain intracellular pH and ion homeostasis and that this is needed for acrosome biogenesis.

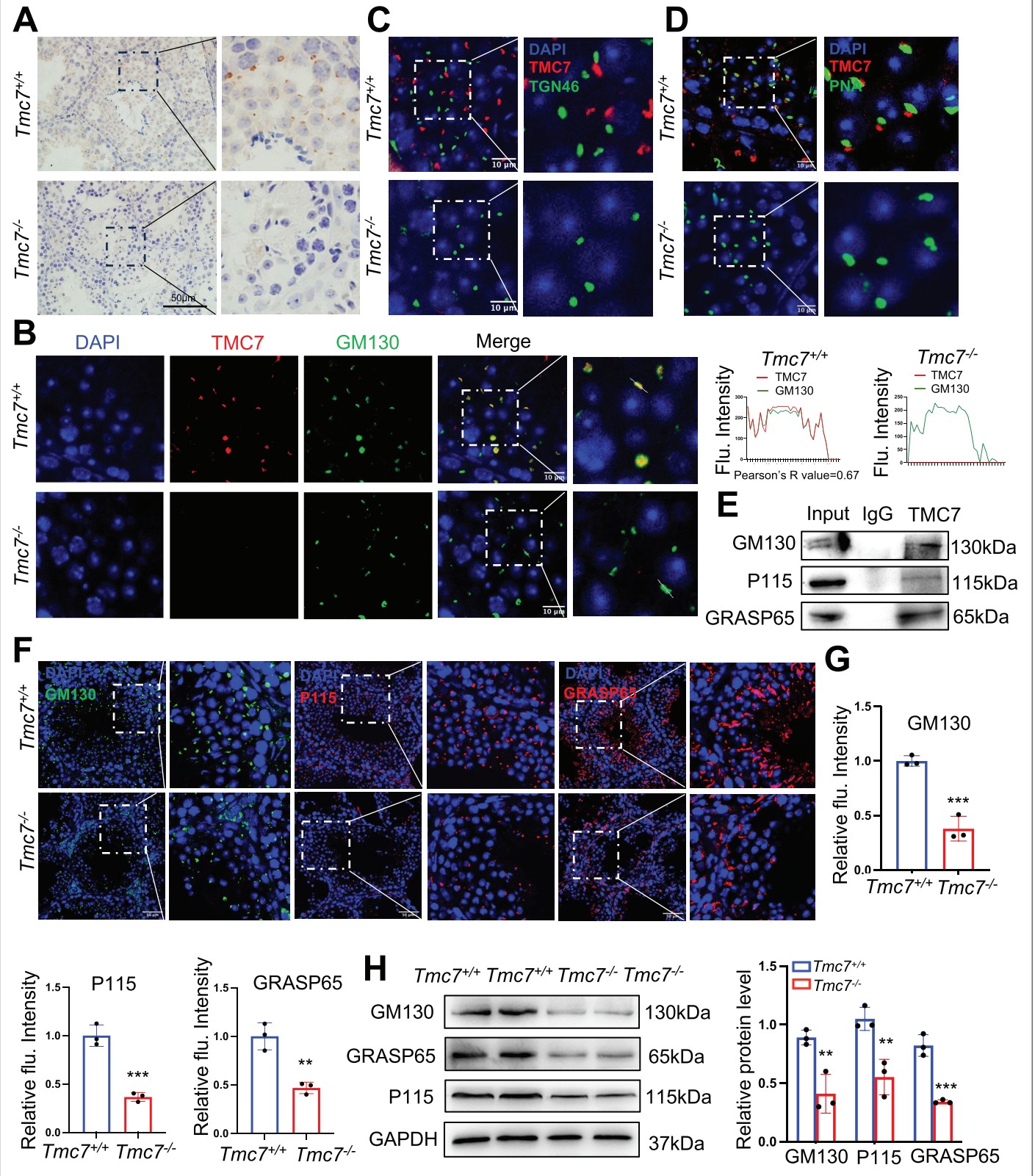

**Figure 4.** TMC7 localizes to the *cis*-Golgi, and this is required for maintaining Golgi integrity. (**A**) Immunohistochemical staining with an antibody against TMC7 in seminiferous tubules of 9-week-old mouse testes. Scale bar, 50 μm. (**B**) Immunofluorescence staining of TMC7 (red) and GM130 (green, a marker of the *cis*-Golgi), with DAPI indicating the nuclei in seminiferous tubules of 9-week-old WT and *Tmc7*⁻/⁻ testis. Scale bar, 10 μm. Colocalization was quantified by ImageJ, and 20 cells were analyzed for the Pearson's R-value. (**C**) Immunofluorescence staining of TMC7 (red) and TGN46 (green, a

*Figure 4 continued on next page*

*Figure 4 continued*

marker of the *trans*-Golgi), with DAPI indicating the nuclei in the seminiferous tubule. Scale bar, 10 µm. (**D**) Immunofluorescence staining of TMC7 (red) and PNA (green, a marker of the acrosome), with DAPI indicating the nuclei in the seminiferous tubule. Scale bar, 10 µm. (**E**) Western blot was used to analyze the results of co-immunoprecipitation and showed the interaction between TMC7 and GM130, P115, and GRASP65 in 9-week-old WT testes. (**F, G**) Immunofluorescence staining and quantification of GM130 (green), P115 (red), and GRASP65 (red) in testicular sections (II–III stage) of 9-week-old WT and *Tmc7⁻/⁻* mice. Scale bar, 50 µm. (**H**) Western blots of the protein levels of GM130 and the GM130-interacting proteins P115 and GRASP65 in 3-week-old WT and *Tmc7⁻/⁻* testes. ImageJ was used to quantify the protein level. Images are representative of at least three independent experiments. The results are presented as the mean ± S.D., and two-sided unpaired Student's *t*-tests were used to calculate the p-values (*p<0.05, **p<0.01, ***p<0.001).

The online version of this article includes the following source data and figure supplement(s) for figure 4:

**Source data 1.** Uncropped and labeled gels for *Figure 4*.

**Source data 2.** Raw unedited gels for *Figure 4*.

**Figure supplement 1.** The location of TMC7 in testis.

**Figure supplement 1—source data 1.** Uncropped and labeled gels for *Figure 4—figure supplement 1*.

**Figure supplement 1—source data 2.** Raw unedited gels for *Figure 4—figure supplement 1*.

## Discussion

Among the eight TMC protein members, TMC1 and TMC2 are localized to the site of MET channels at the tips of the shorter stereocilia of cochlear hair cells (*Beurg et al., 2019*; *Kurima et al., 2015*). However, TMC1 and TMC2 expressed in heterologous cell lines remain in the ER and fail to localize to the plasma membrane (*Nist-Lund et al., 2019a*). TMC4 is a voltage-dependent chloride channel that is predominantly expressed in the taste buds in the posterior region of the tongue (*Kasahara et al., 2021*). TMC6 and TMC8 are localized to the ER and play crucial roles in zinc transport (*Lazarczyk et al., 2008*). However, the physiological function of TMC7 has not been defined. In this study, we demonstrated that *Tmc7* was predominantly expressed in testes (*Figure 1E*), and by generating knockout mice lacking TMC7 we found that TMC7 deficiency resulted in complete male infertility with an OAT-like phenotype. *Tmc7⁻/⁻* mice showed various defects, such as abnormal sperm head, disorganized mitochondrial sheaths, and reduced number of elongating spermatids. Interestingly, we found that TMC7 colocalized with GM130, but not with TGN46 (*Figure 4B and C*), as illustrated by the immunofluorescence co-staining assay, indicating that TMC7 located on the *cis* side of the Golgi apparatus.

Our results demonstrate the critical role of TMC7 during the formation of the acrosome. The acrosome is thought to be derived from the Golgi apparatus, so we speculated that TMC7 deficiency might cause defects in acrosome formation because TMC7 localized on the Golgi apparatus. We observed fragmented and malformed acrosomes and proacrosomal vesicles were found to accumulate around the acrosomes (*Figure 3B*). Similarly, TMC1 deficiency leads to membrane blebbing and PS externalization at the apical region of hair cells. TMC1 is not only an ion channel but also a lipid scramblase, which is required to maintain homeostasis in hair cells (*Ballesteros and Swartz, 2022*). TMC7 has similar amino acid sequence and predicted structure with TMC1/2 (*Figure 5—figure supplement 2*), suggesting TMC7 might function as a lipid scramblase and an ion channel. Further investigations are required to verify the potential functions of TMC7 as a lipid scramblase and an ion channel in spermatid.

In accordance with our findings, other proteins that localize on the Golgi apparatus have also been reported to be involved in acrosome biogenesis. GM130 deficiency results in acrosome biogenesis failure and male infertility because proacrosomal vesicles are unable to fuse to form acrosomes (*Lazarczyk et al., 2008*). Our findings showed that TMC7 was colocalized with GM130 and that TMC7 depletion significantly reduced the protein level of GM130. Importantly, colocalized GOPC and PNA spermatids were significantly decreased in *Tmc7⁻/⁻* spermatids compared to WT spermatids. Further, GOLGA3, PICK1, and several other Golgi proteins are also required for acrosome biogenesis (*Xiong et al., 2021*). Our work, combined with previous studies, broadens our understanding of role the Golgi apparatus plays in the formation of acrosomes.

Our findings highlight the role of TMC7 in maintaining cellular homeostasis. Upon TMC7 depletion, the Golgi apparatus undergoes significant disorganization and is unable to correctly orient toward the acrosome. The mechanism behind this is that intracellular $Ca^{2+}$ concentrations decreased while the pH value was increased in *Tmc7⁻/⁻* germ cells compared to WT germ cells, suggesting that TMC7 is

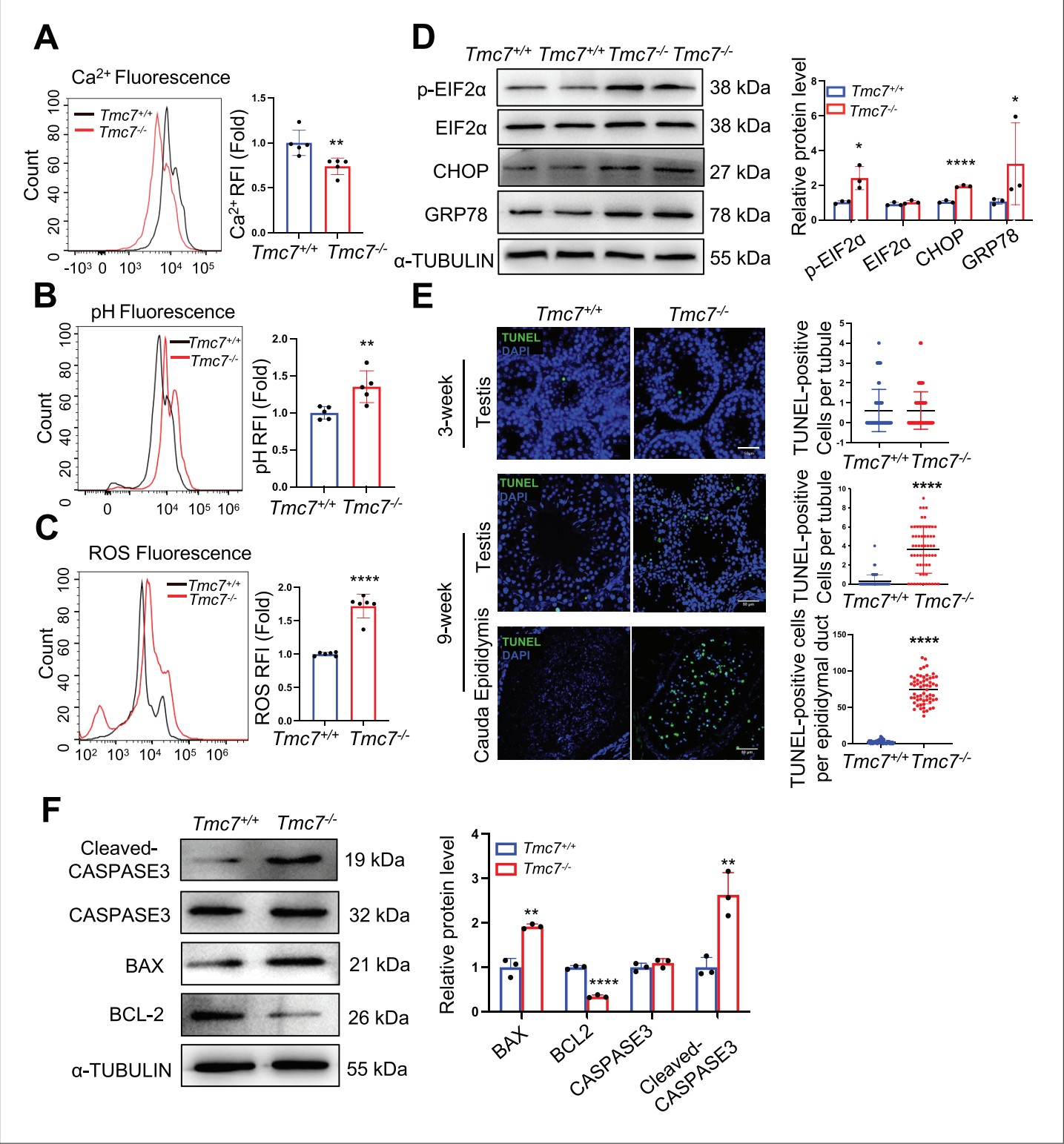

**Figure 5.** TMC7 depletion impairs cellular homeostasis and leads to spermatid apoptosis. (**A**) The $Ca^{2+}$ levels were measured by flow cytometry in WT and $Tmc7^{-/-}$ germ cells isolated from 3-week-old testes. Five mice were used. (**B**) The pH levels were measured by flow cytometry in WT and $Tmc7^{-/-}$ germ cells isolated from 3-week-old testes. Five mice were used. (**C**) The reactive oxygen species (ROS) levels were measured by flow cytometry in WT and $Tmc7^{-/-}$ germ cells isolated from 3-week-old testes. Six mice were used. (**D**) Western blot analysis of protein levels of endoplasmic reticulum (ER) chaperone-associated proteins (p-EIF2α, EIF2α, CHOP, and GRP78) in 3-week-old WT and $Tmc7^{-/-}$ testes. ImageJ was used to quantify the protein level. (**E**) TUNEL assay and the number of TUNEL-positive cells per tubule in seminiferous tubules and cauda epididymis of 3- and 9-week-old WT and $Tmc7^{-/-}$

*Figure 5 continued on next page*

*Figure 5 continued*

male mice. Scale bar, 50 μm. N = 60, 20 tubules in each testis and cauda epididymis were counted, and three mice were used. (**F**) Apoptosis-related proteins (Cleaved-CASPASE3, CASPASE3, BAX, BCL2) were detected by western blotting in 9-week-old WT and $Tmc7^{-/-}$ testes. ImageJ was used to quantify the protein level. Images are representative of at least three independent experiments. The results are presented as the mean ± S.D., and two-sided unpaired Student's *t*-tests were used to calculate the p-values (*p<0.05, **p<0.01, ***p<0.001).

The online version of this article includes the following source data and figure supplement(s) for figure 5:

**Source data 1.** Uncropped and labeled gels for *Figure 5*.

**Source data 2.** Raw unedited gels for *Figure 5*.

**Figure supplement 1.** Intracellular homeostasis was impaired in $Tmc7^{-/-}$ testis.

**Figure supplement 1—source data 1.** Uncropped and labeled gels for *Figure 5—figure supplement 1*.

**Figure supplement 1—source data 2.** Raw unedited gels for *Figure 5—figure supplement 1*.

**Figure supplement 2.** Structure comparison of TMC1/2 with TMC7.

essential for maintaining the intracellular pH and ion homeostasis. Our findings further revealed that TMC7 deficiency led to increased ROS levels, Golgi and ER stress, ultimately resulting in apoptosis of spermatids. The $Na^+/H^+$ exchanger NHE8 is involved in the maintenance of intracellular pH and ion homeostasis, and NHE8 deficiency leads to Golgi-derived vesicles failing to form large acrosomal granules and the acrosomal cap (*Oberheide et al., 2017*). SLC9A3 is another $Na^+/H^+$ exchanger, and it is expressed in the acrosomal region of spermatids and loss of SLC9A3 causes severe acrosomal defects (*Wang et al., 2017*). These studies, together with our current findings, support the hypothesis that TMC7 is essential for maintaining the intracellular pH and ion homeostasis, which is needed for the proper progression of acrosome biogenesis.

In conclusion, our findings show that TMC7 localizes to the *cis*-Golgi region in spermatids and that this is essential for acrosome biogenesis and for male fertility. In the absence of TMC7, intracellular pH and ion homeostasis is disrupted, and this leads to Golgi disorganization and failure to correctly orientate to the acrosome. Subsequently, Golgi-derived proacrosomal vesicles are unable to fuse with the developing acrosome and acrosome biogenesis is severely impaired, and thus TMC7-deficent male mice are completely infertile. Our study highlights the physiological role of TMC family members in spermatogenesis.

## Materials and methods

### Generation of *Tmc7* knockout mice

*Tmc7* conventional knockout mice in the C57BL/6J background were obtained from Cyagen Bioscience (China). Exon 2 was selected as the target site, and Cas9 mRNA and single-guide RNA were co-injected into fertilized eggs for knockout mouse production. The pups were genotyped by PCR using the following primers: F1: 5'-AGA TGG CTT AGT GGG TGC CTT AGT-3'; R1: 5'TGA TGG ACA GAG AGA TGA TGA ATG G-3'; F3: 5'-CAG TCC CAG GTT CAG TTC AGT GAG-3'. In WT mice, the primer pair amplified a 513 bp PCR product, while in $Tmc7^{-/-}$ mice it amplified an 891 bp product. Mice were housed under a controlled illumination regime (light for 12 hr) at 21–22°C and 60% ± 10% relative humidity. Food and water were freely available, and all animal experiments were conducted in accordance with the guidelines of the Animal Care and Use Committee at Shandong University (ECSBMSSDU2022-2-113).

### Metabolic assessment in mice

A Columbus Instruments Comprehensive Lab Animal Monitoring System CLAMS6 was used for the experiment. Nine-week-old WT and $Tmc7^{-/-}$ male mice (n = 4) were weighed and had the metabolic rates measured over a period of 48 hr. The rate of oxygen consumption, RER (breathing entropy), energy expenditure, and food intake were automatically recorded every 8 min. These data were analyzed with a 12 hr dark:12 hr light cycle at 22–23°C.

### RNA isolation and PCR

The tissues were homogenized in TRIzol (Invitrogen, 15596-026) with a grinder. Following centrifugation, the RNA in the supernatant was obtained by precipitating with isopropanol and washing with

ethanol. A total of 1 μg RNA was reverse-transcribed into cDNA with HiScript III RT SuperMix for RT-PCR (+gDNA wiper) (Vazyme, R223-01) according to the manufacturer's instructions. The cDNA was used to detect the expression of Tmc7 with Taq Master Mix (Vazyme, P112-03) with the following primers: F – AGC CAA TCC ATC CGC AAG TAT; R – TGC TGC TG TAC CGT TTC CAC. The products were analyzed by agarose gel electrophoresis.

## Sperm collection

The cauda epididymis was collected and placed in an EP tube containingPhosphate buffer saline (PBS). Sperm were released for 30 min in an atmosphere of 5% $CO_2$ at 37°C. The suspension was diluted at 1:50 and transferred to a hemocytometer for counting. For the anther experiments, the PBS with sperm was spread on the slide and fixed with 4% paraformaldehyde (Servicebio, G1101). The slides were stored at –80°C after drying.

## Sperm motility assessment

For each genotype, at least three mice were used for the sperm motility assessment. 10 μL sperm suspension was added to the 80 mm glass cell chamber (Hamilton Thorne, 80 μm). More than 200 sperm were analyzed through Animal Motility system (Hamilton Thorne, Beverly, MA), the analysis included the total motility, average path velocity, straight-line velocity, and linear velocity. Each analysis was replicated at least five times.

## Histological analysis

For the morphology analysis of the sperm from the caudal epididymis, sperm smears were directly stained with hematoxylin and sealed with neutral balsam. The tissue was fixed in 4% paraformaldehyde overnight, then dehydrated and embedded in paraffin. Paraffin-embedded tissues were sectioned at 4 μm. Tissue slides were deparaffinized in xylene and dehydrated in a series of ethanol. The tissues were then stained by hematoxylin or with periodic acid staining using a PAS Stain Kit (Abcam, ab150680) according to the manufacturer's instructions. Finally, tissue slides were rehydrated in a series of ethanol and xylene and sealed with neutral balsam. An Olympus BX53 microscope system was used for imaging.

## Immunofluorescence

We performed immunofluorescence of germ cells in the testes using two methods, including testis slides and squashed seminiferous tubules. The squashed seminiferous tubules were prepared according to a previous report (*Wellard et al., 2018*). The testis slides were deparaffinized in xylene and rehydrated in a series of ethanol. Antigen retrieval was performed by heating in EDTA Antigen Retrieval Solution (pH 9.0) (ZSGB-BIO, ZLI-9096), and nonspecific binding sites were blocked with EZ-Buffers M 1X Block BSA in PBS (Sangon, C520035-0250). The slides were incubated with primary antibody overnight at 4°C. Primary antibodies included anti-TMC7 (1:200 dilution, Abcam, ab191521), anti-SOX9 (1:500 dilution, Sigma, AB5535), anti-GOPC (1:200 dilution, Proteintech, 12163-1-AP), anti-SYCP3 (1:500 dilution, Santa Cruz, sc-33195), anti-H2AX (1:500 dilution, Millipore, 05-636-1), anti-GM130 (1:100 dilution, 610822, BD Biosciences), anti-P115 (1:200 dilution, Proteintech, 13509-1-AP), and anti-GRASP65 (1:200 dilution, Proteintech, 10747-2-AP). The next day they were incubated with the secondary antibody (1:500 dilution, Thermo, A-11070 and A-11012) for 1 hr at room temperature. The Alexa Fluor 488 conjugates of lectin PNA (1:500 dilution, Thermo, L21409), anti-GM130 (1:200 dilution, Proteintech, CL488-11308), and anti-TGN46 (1:200 dilution, Proteintech, CL488-13573) were used as secondary antibodies. For the staining of Mito-track (1:5000 dilution, Beyotime, C1048), the sperm released from the caudal epididymis were directly stained for 30 min at 37°C. The sperm were then spread on the slide and observed. A TUNEL assay was performed using a one-step TUNEL Apoptosis Assay Kit (KeyGEN BioTECH, KGA7072) according to the manufacturer's instructions. Briefly, the slide was deparaffinized and antigen retrieval was performed. dUTP was used to label the cells, and the nuclei were stained with DAPI (Abcam, ab285390). Fluorescent images were captured on a Dragonfly 200 confocal microscopy system (Andor Technology).

## Western blotting

The membrane and cytoplasmic proteins of the testis were obtained using a membrane protein extraction kit (Abmart, A1008) according to the manufacturer's instructions. Total protein in the testis was obtained by grinding in a tissue grinder in RIPA lysis buffer (Beyotime, P0013B), and the concentration of protein was measured using a BCA protein assay kit (Beyotime, P0009). The proteins were separated by SDS-PAGE and then transferred to a Polyvinylidene fluoride (PVDF) membrane. The membranes were blocked in 5% non-fat milk and then incubated with primary antibodies overnight at 4°C. Primary antibodies included anti-TMC7 (Dia-an Biotech), anti-ATP1A1 (Proteintech, 14418-1-AP), anti-GAPDH (Proteintech, 60004-1-1g), anti-Cleaved-CASPASE3 (Cell Signaling Technology, 9661), anti-CASPASE3 (Cell Signaling Technology, 9662), anti-BAX (Cell Signaling Technology, 41162), anti-BCL2 (Cell Signaling Technology, 17447), anti-α-TUBULIN (Proteintech, 11224-1-AP), anti-P-EIF2α (ABclonal, AP0745), anti-EIF2α (ABclonal, A0764), anti-GRP78 (Proteintech, 11587-1-AP), anti-CHOP (Proteintech, 15204-1-AP), anti-P115 (Proteintech, 13509-1-AP), and anti-GRASP65 (Proteintech, 10747-2-AP). All antibodies were diluted 1:1000. The next day, the membrane was incubated with horseradish peroxidase-conjugated secondary antibody for 1 hr, and the protein was detected using an ECL system (Vazyme, E412-02). The quantitation was performed in Image J.

## Chromosome spreading

Meiotic nuclear spreading was performed as previously reported (*Peters et al., 1997*). Briefly, spermatocytes released from seminiferous tubules were incubated in hypotonic solution for 1 hr and then transferred to 100 mM sucrose. The suspension was spread on the glass slides with 1% PFA and 0.15% Triton X100, and dried slides were stored at –80°C or were used for the immunofluorescence staining.

## Transmission electron microscopy

The testes of 9-week-old WT and *Tmc7*$^{-/-}$ mice (n = 3) were used for the TEM analysis. The testes were washed in PBS three times and then fixed in 1% $O_sO_4$ in 0.1 M PBS (pH 7.4) buffer at 4°C overnight. The testes were then fixed in 1% $O_sO_4$ at room temperature for 1 hr. The samples were then washed with water three times for 15 min each and dehydrated in a graded ethanol series (50%, 70%, 95%, and 100%). The samples were finally embedded with resin, cut into 80 nm sections, and viewed with a JEM-1400 transmission electron microscope.

## Flow cytometry

The testes of PD21 and PD30 WT and *Tmc7*$^{-/-}$ mice (n = 3–6) were used to obtain germ cells. The tunica albuginea was removed and placed into PBS with 120 U/ml Type I Collagenase (Gibco, 17100-017) on a shaker at 37°C and 220 rpm for 10 min. The supernatant was filtered through a 70 µm cell filter screen into a new tube. The sediment was digested again with 0.25% trypsin-EDTA (Gibco, 25200-072) on a shaker at 37°C and 220 rpm for 10 min and then neutralized by serum albumin and filtered. The germ cells were obtained by centrifuging twice at 1000 × *g* for 5 min and then removing the supernatant.

The levels of $Ca^{2+}$ (Beyotime, Fluo-3 AM, S1056), pH (Beyotime, BCECF AM, S1006), and ROS (Beyotime, DCFH-DA, S0033S) were measured using fluorescent probes according to the manufacturer's protocols. The probes were all diluted 1:2000. Germ cells were stained with the probes at 37°C for 30 min, then washed with PBS three times. The supernatant was filtered through a 45 µm cell filter screen and analyzed by flow cytometry (Gallios, USA).

## Co-immunoprecipitation

Testes of 9-week-old WT mice were ground in a tissue grinder in RIPA Lysis Buffer (Beyotime, P0013B) to obtain the total protein. TMC7 (Dia-an Biotech) and IgG (Beyotime, A7016) antibodies were incubated with the proteins overnight. The next day, they were incubated with protein A/G magnetic beads (MCE, HY-K0202) for 2 hr. The beads were wash and boiled for 10 min in 1× SDS loading buffer. Western blotting was used for the analysis of input and immunoprecipitated samples with anti-P115 and anti-GRASP65 antibodies.

## Statistical analysis

GraphPad Prism 9 was used to analyze the data, and the results are shown as the mean ± standard deviation (S.D.). Statistical significance was determined using two-sided Student's *t*-tests or two-way ANOVA. *p<0.05, **p<0.01, and ***p<0.001 indicate the level of statistical significance, and ns indicates not significant.

## Acknowledgements

We thank the Translational Medicine Core Facility of Shandong University for consultation and instrument availability that supported this work. We also thank the Laboratory Animal Center of Shandong University for mouse housing and care. This work was supported by the National Natural Science Foundation of China (82172718, 82301801, 82303055, 81972437).

## Additional information

### Funding

| Funder | Grant reference number | Author |
| --- | --- | --- |
| National Natural Science Foundation of China | 82172718 | Chunhong Qiu |
| National Natural Science Foundation of China | 82301801 | Junchao Qin |
| National Natural Science Foundation of China | 82303055 | Zixiang Wang |
| National Natural Science Foundation of China | 81972437 | Zhaojian Liu |
| Natural Science Fundation of Shandong Province | ZR2023MC067 | Xiyu Zhang |

The funders had no role in study design, data collection and interpretation, or the decision to submit the work for publication.

### Author contributions

Jing Wang, Data curation, Validation, Methodology, Writing – original draft; Yingying Yin, Data curation, Validation, Investigation, Methodology; Lei Yang, Data curation, Software, Investigation, Methodology; Junchao Qin, Resources, Validation, Methodology; Zixiang Wang, Software, Formal analysis, Methodology; Chunhong Qiu, Resources, Methodology, Writing – review and editing; Yuan Gao, Resources, Supervision, Writing – review and editing; Gang Lu, Supervision, Writing – review and editing; Fei Gao, Conceptualization, Writing – review and editing; Zi-Jiang Chen, Resources, Writing – review and editing; Xiyu Zhang, Writing – original draft, Project administration, Writing – review and editing; Hongbin Liu, Resources, Writing – original draft, Project administration, Writing – review and editing; Zhaojian Liu, Supervision, Writing – original draft, Project administration, Writing – review and editing

### Author ORCIDs

Jing Wang http://orcid.org/0000-0002-5098-6619
Zixiang Wang https://orcid.org/0000-0001-5252-9764
Fei Gao https://orcid.org/0000-0002-4029-6411
Zi-Jiang Chen https://orcid.org/0000-0001-6637-6631
Hongbin Liu https://orcid.org/0000-0003-2550-7492
Zhaojian Liu https://orcid.org/0000-0002-2542-0859

### Ethics

All animal experiments were conducted in accordance with the guidelines of the Animal Care and Use Committee at Shandong University (ECSBMSSDU2022-2-113).

Reviewer #1 (Public Review): https://doi.org/10.7554/eLife.95888.4.sa1
Reviewer #2 (Public Review): https://doi.org/10.7554/eLife.95888.4.sa2
Reviewer #3 (Public Review): https://doi.org/10.7554/eLife.95888.4.sa3
Author response https://doi.org/10.7554/eLife.95888.4.sa4

## Additional files

### Supplementary files

- Supplementary file 1. Fertility testing of *Tmc7*[+/-], *Tmc7*[-/-] male mice and *Tmc7*[-/-] female mice.
- MDAR checklist

### Data availability

All data generated or analysed during this study are included in the manuscript and supporting files.

The following previously published dataset was used:

| Author(s) | Year | Dataset title | Dataset URL | Database and Identifier |
|-----------|------|---------------|-------------|-------------------------|
| Moreira MC | 2019 | Mouse RNA-seq time-series of the development of seven major organs | https://www.ebi.ac.uk/biostudies/arrayexpress/studies/E-MTAB-6798 | ArrayExpress, E-MTAB-6798 |

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
