## [Editor Report · eLife assessment]

This study reports an **important** discovery highlighting the essential role of the putative ion channel, TMC7, in acrosome formation during sperm development and thus male fertility. The evidence for the requirement of TMC7 in acrosome biogenesis and sperm function is **convincing**, although its function as an ion channel remains to be further determined. Overall, this work will be of great interest to developmental biologists and ion channel physiologists alike.

---

## [Referee Report · Reviewer #1 (Public Review)]

Summary:

TMC7 knockout mice were generated by the authors and the phenotype was analyzed. They found that Tmc7 is localized to Golgi and is needed for acrosome biogenesis.

Strengths:

The phenotype of infertility is clear, and the results of TMC7 localization and the failed acrosome formation are highly reliable. In this respect, they made a significant discovery regarding spermatogenesis.

In the original version, I pointed out the gap between their pH/calcium imaging data and the hypothesis of ion channel function of TMC7 in the Golgi. Now the author agrees and has changed the description to be reasonable. Additional experiments were also performed, and I can say that they have answered my concern adequately.

I would say it is good to add any presumed mechanism for the observed changes in pH and calcium concentration in the cytoplasm this time.

---

## [Referee Report · Reviewer #2 (Public Review)]

Summary:

This study presents a significant finding that enhances our understanding of spermatogenesis. TMC7 belongs to a family of transmembrane channel-like proteins (TMC1-8), primarily known for their role in the ear. Mutations to TMC1/2 are linked to deafness in humans and mice and were originally characterized as auditory mechanosensitive ion channels. However, the function of the other TMC family members remains poorly characterized. In this study, the authors begin to elucidate the function of TMC7 in acrosome biogenesis during spermatogenesis. Through analysis of transcriptomics datasets, they elevated levels of TMC7 in round spermatids in both mouse and human testis. They then generate Tmc7-/- mice and find that male mice exhibit smaller testes and complete infertility. Examination of different developmental stages reveals spermatogenesis defects, including with reduced sperm count, elongated spermatids and large vacuoles. Additionally, abnormal acrosome morphology are observed beginning at the early-stage Golgi phase, indicating TMC7's involvement in proacrosomal vesicle trafficking and fusion. They observed localization of TMC7 in the cis-Golgi and suggest that its presence is required for maintaining Golgi integrity, with Tmc7-/- leading to reduced intracellular Ca2+, elevated pH and increased ROS levels, likely resulting in spermatid apoptosis. Overall, the work delineates a new function of TMC7 in spermatogenesis and the authors propose that its ion channel and/or scramblase activity is likely important for Golgi homeostasis. This work is of significant interest to the community and is of high quality.

Strengths:

The biggest strength of the paper is the phenotypic characterization of the TMC7-/- mouse model, which has clear acrosome biogenesis/spermatogenesis defects. This is the main claim of the paper and it is supported with the data that are presented.

Weaknesses:

It isn't clear whether TMC7 functions as an ion channel from the current data presented in this paper, but the authors are careful in their interpretation and present this merely as a hypothesis supporting this idea.

---

## [Referee Report · Reviewer #3 (Public Review)]

Summary:

In this study, Wang et al. have demonstrated that TMC7, a testis-enriched multipass transmembrane protein, is essential for male reproduction in mice. Tmc7 KO male mice are sterile due to reduced sperm count and abnormal sperm morphology. TMC7 co-localizes with GM130, a cis-Golgi marker, in round spermatids. The absence of TMC7 results in reduced levels of Golgi proteins, elevated abundance of ER stress markers, as well as changes of Ca2+ and pH levels in the KO testis. However, further confirmation is required because the analyses were performed with whole testis samples in spite of the differences in the germ cell composition in WT and KO testis. In addition, the causal relationships between the reported anomalies await thorough interrogation

Strengths:

By using PD21 testes, the revised assays have consolidated that depletion of TMC7 leads to a reduced level of Ca2+ and an elevated level of ROS in the male germ cells. The immunohistochemistry analyses have clearly indicated the reduced abundance of GM130, P115, and GRASP65 in the knockout testis.

Weaknesses:

Future studies are required to decipher how TMC7 stabilizes Golgi structure, coordinates vesicle transport, and maintains the germ cell homeostasis.

---

## [Author Response]

The following is the authors’ response to the previous reviews.

**Public Reviews:**

**Reviewer #1 (Public Review):**
Summary:TMC7 knockout mice were generated by the authors and the phenotype was analyzed. They found that Tmc7 is localized to Golgi and is needed for acrosome biogenesis.Strengths:The phenotype of infertility is clear, and the results of TMC7 localization and the failed acrosome formation are highly reliable. In this respect, they made a significant discovery regarding spermatogenesis.In the original version, I pointed out the gap between their pH/calcium imaging data and the hypothesis of ion channel function of TMC7 in the Golgi. Now the author agrees and has changed the description to be reasonable. Additional experiments were also performed, and I can say that they have answered my concern adequately.I would say it is good to add any presumed mechanism for the observed changes in pH and calcium concentration in the cytoplasm this time.

We appreciate your positive comments on our revised manuscript.

**Reviewer #2 (Public Review):**
Summary:This study presents a significant finding that enhances our understanding of spermatogenesis. TMC7 belongs to a family of transmembrane channel-like proteins (TMC1-8), primarily known for their role in the ear. Mutations to TMC1/2 are linked to deafness in humans and mice and were originally characterized as auditory mechanosensitive ion channels. However, the function of the other TMC family members remains poorly characterized. In this study, the authors begin to elucidate the function of TMC7 in acrosome biogenesis during spermatogenesis. Through analysis of transcriptomics datasets, they identify TMC7 as a transmembrane channel-like protein with elevated transcript levels in round spermatids in both mouse and human testis. They then generate Tmc7-/- mice and find that male mice exhibit smaller testes and complete infertility. Examination of different developmental stages reveals spermatogenesis defects, including reduced sperm count, elongated spermatids, and large vacuoles. Additionally, abnormal acrosome morphology is observed beginning at the early-stage Golgi phase, indicating TMC7's involvement in proacrosomal vesicle trafficking and fusion. They observed localization of TMC7 in the cis-Golgi and suggest that its presence is required for maintaining Golgi integrity, with Tmc7-/- leading to reduced intracellular Ca2+, elevated pH, and increased ROS levels, likely resulting in spermatid apoptosis. Overall, the work delineates a new function of TMC7 in spermatogenesis and the authors suggest that its ion channel activity is likely important for Golgi homeostasis. This work is of significant interest to the community and is of high quality.Strengths:The biggest strength of the paper is the phenotypic characterization of the TMC7-/- mouse model, which has clear acrosome biogenesis/spermatogenesis defects. This is the main claim of the paper and it is supported by the data that are presented.Weaknesses:The claim is that TMC7 functions as an ion channel. It is reasonable to assume this given what has been previously published on the more well-characterized TMCs (TMC1/2), but the data supporting this is preliminary here, and more needs to be done to solidify this hypothesis. The authors are careful in their interpretation and present this merely as a hypothesis supporting this idea.

We appreciate this constructive suggestion.

**Reviewer #3 (Public Review):**
Summary:In this study, Wang et al. have demonstrated that TMC7, a testis-enriched multipass transmembrane protein, is essential for male reproduction in mice. Tmc7 KO male mice are sterile due to reduced sperm count and abnormal sperm morphology. TMC7 co-localizes with GM130, a cis-Golgi marker, in round spermatids. The absence of TMC7 results in reduced levels of Golgi proteins, elevated abundance of ER stress markers, as well as changes of Ca2+ and pH levels in the KO testis. However, further confirmation is required because the analyses were performed with whole testis samples in spite of the differences in the germ cell composition in WT and KO testis. In addition, the causal relationships between the reported anomalies await thorough interrogationStrengths:By using PD21 testes, the revised assays have consolidated that depletion of TMC7 leads to a reduced level of Ca2+ and an elevated level of ROS in the male germ cells. The immunohistochemistry analyses have clearly indicated the reduced abundance of GM130, P115, and GRASP65 in the knockout testis.Weaknesses:The Discussion section contains sentences reiterating the Introduction and Results of this manuscript (e.g., Lines 79-85 and 231-236; Lines 175-179 and 259-263). Those read repetitive and can be removed.

We thank the reviewer for this import comment. We have modified the text according to your suggestion.

Future studies are required to decipher how TMC7 stabilizes Golgi structure, coordinates vesicle transport, and maintains the germ cell homeostasis.

Thanks. We appreciate this constructive suggestion. We totally agree the reviewer that future studies are required to decipher how TMC7 stabilizes Golgi structure, coordinates vesicle transport, and maintains the germ cell homeostasis.

**Recommendations for the authors**

**Reviewer #1 (Recommendations For The Authors):**
1. In Fig S6d, the bar of Tmc7-/- is broken in the middle for P-EIF2.

Thanks. We have remade Fig S6d according to your suggestion in the revised manuscript.

**Reviewer #2 (Recommendations For The Authors):**
None. The reviewers have adequately answered my points. Many thanks!

We thank the reviewer for accepting our revisions as sufficient.

**Reviewer #3 (Recommendations For The Authors):**
In the revised manuscript, the authors have addressed most of my concerns.

We are pleased that we were able to adequately address the reviewer’s concerns. We appreciate your suggestions to further improve our study.